# Cardiovascular Risk Factors, Angiographical Features and Short-Term Prognosis of Acute Coronary Syndrome in People Living with Human Immunodeficiency Virus: Results of a Retrospective Observational Multicentric Romanian Study

**DOI:** 10.3390/diagnostics13091526

**Published:** 2023-04-24

**Authors:** Mircea Bajdechi, Adriana Gurghean, Vlad Bataila, Alexandru Scafa-Udriste, Roxana Radoi, Anca Cristiana Oprea, Adrian Marinescu, Stefan Ion, Valentin Chioncel, Alina Nicula, Achilleas Anastasiou, Georgiana-Elena Bajdechi, Ilinca Savulescu-Fiedler, Irina Magdalena Dumitru, Sorin Rugina

**Affiliations:** 1Faculty of Medicine, University of Medicine and Pharmacy “Carol Davila” of Bucharest, 050474 Bucharest, Romania; mircea.bajdechi@gmail.com (M.B.); alexandru.scafa@umfcd.ro (A.S.-U.); cristiana_oprea@umfcd.ro (A.C.O.); valentin.chioncel@umfcd.ro (V.C.); alina.nicula@umfcd.ro (A.N.); georgianastoian1993@gmail.com (G.-E.B.); ilincasavulescu@gmail.com (I.S.-F.); 2Doctoral School of Medicine, “Ovidius” University of Constanta, 900470 Constanta, Romania; dumitrui@hotmail.com (I.M.D.); sorinrugina@yahoo.com (S.R.); 3Emergency Clinical Hospital of Bucharest, 014461 Bucharest, Romania; vladbataila1@gmail.com; 4Clinical Hospital of Infectious and Tropical Disease “Dr. Victor Babes” of Bucharest, 030303 Bucharest, Romania; 5National Institute of Infectious Disease “Prof. Dr. Matei Bals” of Bucharest, 021105 Bucharest, Romania; 6Faculty of Medicine, “Titu Maiorescu” University of Bucharest, 040441 Bucharest, Romania; 7Departament of Statistics and Actuarial-Financial Mathematics, Laboratory of Statistics and Data Analysis, University of Aegean, 83200 Samos, Greece; sasd200003@sas.aegean.gr; 8Clinical Infectious Disease Hospital of Constanta, 900178 Constanta, Romania; 9Romanian Academy of Medical Sciences, 030167 Bucuresti, Romania; 10Academy of Romanian Scientists, 030167 Bucuresti, Romania

**Keywords:** cardiovascular risk factors, angiography, acute coronary syndrome, SYNTAX, people living with human immunodeficiency virus

## Abstract

People living with human immunodeficiency virus have increased cardiovascular risk due to a higher prevalence of traditional and particular risk factors such as chronic inflammation, immune dysregulation, endothelial dysfunction, coagulation abnormalities and antiretroviral therapy. In developed countries, coronary artery disease has become the most frequent cardiovascular disease and an important cause of mortality in these patients. The symptomatology of an acute coronary syndrome can be atypical, and the prevalence of each type of acute coronary syndrome is reported differently. Regarding coronary artery disease severity in people living with HIV, the literature data indicates that the presence of single-vessel disease is akin to that of HIV-negative patients, and their short-term prognosis is unclear. This study aims to assess the clinical characteristics, biological parameters, angiographical features and short-term prognosis of acute coronary syndrome in a cohort of Romanian people living with human immunodeficiency virus.

## 1. Introduction

Highly active antiretroviral therapy increases the life expectancy and improves the quality of life of people living with human immunodeficiency virus (PLWH), who have an increased cardiovascular risk compared to the general population. PLWH are twice as likely to develop cardiovascular disease, hence the global burden of HIV-associated cardiovascular disease tripling over the past 26 years [1]. It is anticipated that by 2030, most people living with HIV will be over the age of 50 and 78% of them will be suffering from cardiovascular disease [2]. The mortality rate due to cardiovascular diseases in HIV-infected patients was reported as 6.5% in Europe [3], 8% in France [4] and 15% in the USA [5].

Atherosclerosis is higher and more severe among people living with HIV, with or without antiretroviral treatment, than among the HIV-negative population [6]. As highlited in the literature, the high prevalence of traditional cardiovascular risk factors and specific risk factors such as chronic inflammation, immune dysregulation, endothelial dysfunction and coagulation abnormalities can lead to accelerated and diffuse atherosclerosis in people living with HIV [7,8,9,10,11]. Exposure to certain antiretroviral drugs plays an important role in the exacerbation of cardiovascular risk factors [12,13]. Severe atherosclerosis of the porcelain aorta type was reported in a young patient on long-term antiretroviral therapy, but without being able to establish a cause [14]. There are three key sequential biological processes that accelerate the progression of atherosclerosis in PLWH [6]: 1-inflammation; 2-transformation of monocytes to macrophages and foam cells; 3-Ca^2+^ dependent endoplasmic reticulum stress and apoptosis of foam cells, leading to plaque development. 

Therefore, in the last 10 years, coronary artery disease has become the most impactful cardiovascular disease in people living with HIV in developed countries [5]. These patients have a high risk of acute coronary syndrome [8,15,16,17,18]. Despite the high cardiovascular risk, there are relatively few data on coronary artery disease severity and management among PLWH. Some studies that analyzed acute coronary syndromes according to type found a higher prevalence of ST elevation acute coronary syndrome (STE-ACS) [17], while others showed a higher prevalence of non-STE elevation acute coronary syndrome (NSTE-ACS) [19,20,21] than controls. Regarding the extension of coronary artery disease, the literature data reported similar ubiquity of single-vessel disease to that of HIV-negative patients [20,22,23]. People living with HIV appear to have a higher frequency of left anterior descending artery culprit lesions compared to controls [20,21], but there are studies that reported opposing data [19]. Studies usually did not report significant severe coronary artery disease extension in people living with HIV who presented with acute coronary syndrome, compared to controls: Hsue et al. [22] reported a more extensive disease in HIV-negative patients, Badr et al. [24] showed less complex angiographic lesions in PLWH, Baccaro et al. reported no difference regarding extent and severity of coronary artery disease between the two groups [20]. The assessment of coronary artery disease by initial SYNTAX (SYNergy between percutaneous coronary intervention with TAXus and coronary artery bypass surgery) or Gensini score showed no significant differences between the groups [19,23], whereas PLWH had significantly higher residual SYNTAX scores [23].

Studies using invasive and non-invasive methods have highlighted the particularities of coronary atherosclerotic involvement [11,25]. The analysis of coronary lesions by Coronary Computed Tomography Angiography (CCTA) as well as Intravascular Ultrasound (IVUS) showed a higher prevalence of non-calcified, soft atheroma plaques which have a higher risk of acute coronary syndrome in HIV-infected patients. Aging, hypertension, diabetes mellitus and dyslipidemia hasten the process of coronary sclerosis and calcification. The lower rate of calcified plaques in people living with HIV presenting with acute coronary syndrome reflects the lower burden of traditional risk factors [25]. The intravascular ultrasound plaque examination in PLWH revealed hypoechoic non-calcified plaques, emphasizing the persistent inflammation burden [25].

Even though the risk of acute coronary syndrome recurrence is higher in the short- and long-term follow-up, the other major adverse cardiac and cerebrovascular events (MACCE) seem to have the same prevalence in long-term follow-up in PLWH and HIV-negative cohorts [19,20,26,27]. Boccara et al. showed that after coronary artery bypass graft revascularization, PLWH had higher rates of adverse events compared to controls, mostly caused by the need for repeated revascularization using percutaneous coronary intervention of the native coronary arteries [28].

There are various data about the characteristics, biological parameters and angiographic features of PLWH presenting with acute coronary syndrome and their short-term prognosis. Due to the lack of data in this geographic region, where the overall cardiovascular risk in the general population is higher than in other parts, our interest in defining the differences in ACS in a Romanian HIV cohort has increased. The aim of this study is to assess the main characteristics of a Romanian HIV cohort presenting with ACS and their 30-day follow-up.

## 2. Materials and Methods

### 2.1. Study Design

We conducted a retrospective observational multicentric study to compare characteristics related to acute coronary syndrome in people living with HIV and HIV-negative patients. We used medical databases accessing the main departments of cardiology and infectious diseases from Bucharest and Constanta, Romania, between October 2009 and October 2022. The diagnosis of acute coronary syndrome was established according to the European Society of Cardiology definition and included ST-segment elevation myocardial infarction (STEMI), non-ST-segment elevation myocardial infarction (NSTEMI) and unstable angina (UA). The inclusion criteria were: PLWH diagnosed with acute coronary syndrome, documented HIV infection, age > 18 years. The exclusion criteria were: patient refusal or PLWH diagnosed with endocarditis, pulmonary hypertension or pregnancy. The study was approved by the “Ovidius” University of Constanta Ethic Committee, approval no. 4216129, approval date 29 March 2021.

We identified 50 people living with HIV with ACS (group A), and we compared their characteristics with a control group consisting of 50 consecutive patients without HIV infection with ACS (group B), matched for age and sex.

Patients were analyzed for their cardiovascular risk factors, medical history and therapies. Cardiovascular risk factors at admission included smoking (both current and previous), dyslipidemia, arterial hypertension (blood pressure ≥ 130/80 mmHg or on antihypertensive drugs), diabetes mellitus (fasting hyperglycemia ≥ 126 mg/dL or on antidiabetic treatment), obesity or personal history of coronary artery disease. We noted the type of angina as a clinical parameter. We considered atypical angina to be pain that did not meet all the characteristics of angina or dyspnea as a main symptom at presentation of an ACS. Biological features such as lipidic profile (consisting of total cholesterol, triglycerides), hemoglobin, creatinine, CKD-EPI Equations for Glomerular Filtration Rate, total cholesterol, CK and CK-MB peaks were collected. We identified the culprit lesions and we assessed coronary heart disease extension by SYNTAX I score. The prospective part of this study consists of following the patients longitudinally for incidence of 30-day major adverse cardiac and cerebrovascular events. 

The primary endpoint was a composite of short-term prognoses consisting of in-hospital mortality and 30-day MACCE comprising recurrent ACS, HF requiring hospitalization, cardiac death and stroke.

### 2.2. Statistical Analysis

The database was created in Microsoft Office Professional Plus 2016 (Microsoft Excel), then exported to Statistical Package for the Social Sciences (SPSS) to be processed. Continuous variables are presented as mean (standard deviation—SD). The student’s *t* test and chi^2^ test were used to evaluate statistical significance between categorical and continuous variables. All the differences were considered significant at a two-tailed *p*-value < 0.05. We used logistic regression to analyze the relationship between various independent predictors and MACCE.

## 3. Results

All results related to epidemiological data, clinical characteristics, medical history, comorbidities, diagnosis, angiographical features, biological parameters, SYNTAX scores and short-term prognostic factors are listed in Table 1, Table 2, Table 3, Table 4, Table 5, Table 6 and Table 7. A total of 50 PLWH and 50 HIV-uninfected controls hospitalized with acute coronary syndrome were enrolled. The mean age of the cohort was 49.62 (SD 11.43) years and 94% were men. The control group included HIV-uninfected patients matched for age and sex.

### 3.1. Clinical Features

Cardiovascular risk factors were balanced between the two groups (Table 1); however, PLWH had significantly lower rates of smoking (64% vs. 86%, *p* = 0.011) and obesity (8% vs. 34%, *p* = 0.003). There were no significant differences between other cardiovascular risk factors. PLWH had atypical angina at ACS onset more frequently (20% vs. 2%, *p* = 0.003).

Significant differences were observed in the two groups regarding the types of acute coronary syndrome (Table 2). From PLWH, 42% were admitted to the hospital with STEMI vs. 74% of controls, 24% were diagnosed with NSTEMI vs. 10% of controls and 34% suffered from UA vs. 16% (global *p* = 0.012). Furthermore, PLWH had a lower Killip I severity class than controls (78.8% vs. 90.5%, *p* = 0.058).

### 3.2. Angiographic Features 

There were some differences between culprit vessels in both groups, but without statistical significance (Table 2). The most common was the left anterior descending artery in both groups (46.5% in group A vs. 40% in group B). The culprit vessel was the left circumflex artery in 16.3% of the patients in group A, compared to 24% of the patients in group B; the right coronary artery for 16.3% of the patients in group A and 32% of the patients in group B; and the left main artery in 7% of the patients in group A and 2% of the patients in group B (global *p* = 0.063).

By analyzing the number of diseased vessels, we have found significant differences between the two groups (Table 2). PLWH had lower prevalence of single-vessel disease (40.47% vs. 59%) and two-vessel disease (23.81% vs. 30%), and they had higher prevalence of three-vessel disease (35.72% vs. 12%, global *p* = 0.025).

We analyzed the coronary artery disease extension and severity by calculating SYNTAX I scores, and we observed significant differences (Table 2). A total of 24.3% of group A patients had a SYNTAX I score ≥ 23 points, compared to 4.4% in group B (*p* = 0.019). Furthermore, the differences are significant between the continuous variables of the values of SYNTAX I scores among the two groups (13.45 [SD 11.82] points vs. 7.86 [SD 9.78] points, *p* = 0.022). 

### 3.3. Biological Features

PLWH had lower hemoglobin values than controls, without reaching a statistically significant difference (*p* = 0.141). The values of mean corpuscular volume (MCV) were higher in the PLWH group (mean value of 95.32, [SD 9.92 fL]) than the control group (mean value of 89.19, [SD 9.33 fL]), a result with powerful statistical significance, *p* = 0.002. Serum creatinine values did not differ significantly between the two groups, but patients included in group A had lower glomerular filtration rate values compared to patients included in group B (85.51 [DS 33.51] mL/min/m^2^ vs. 101.79 [DS 31.30] mL/min/m^2^), differences which were statistically significant (*p* = 0.014). Unlike HIV-negative controls, total cholesterol (173.21 [DS 50.79] mg/dL vs. 207.50 [DS 62.16] mg/dL, *p* = 0.004), total CK (715.95 [DS 985.80] U/L vs. 1485.44 [DS 1514.32] U/L, *p* = 0.008) and CK-MB (103.72 [DS 106.40] U/L vs. 157.06 [DS 139.77] U/L, *p* = 0.039) were significantly lower in PLWH (Table 3).

### 3.4. Characteristics of People Living with HIV

We analyzed general features of HIV infections such as the stage of the disease, immune status and antiretroviral therapy. Most of the patients had B1 stage (24%) or C2 stage (20%), and 6% of them had C3 stage. We did not have sufficient data about the stage of the infection in 6% of the patients (Table 4). Regarding the immune status at the time of the onset of the acute coronary syndrome, most patients (54%) had CD4 + lymphocytes over 500 cells/µL, over a third of them had values between 200 and 500 cells/µL and only 6% had below 200 cells/µL (Table 5). Antiretroviral therapy exposure was taken into account, and it was found that 90% of all patients were treated with antiretroviral regimens. The rates of exposure to various antiretroviral classes and types are illustrated in Table 6.

### 3.5. Follow-Up

Short-term follow-up included duration of hospitalization, in-hospital mortality and 30-day MACCE assessment (Table 7). The two groups had no significant differences regarding duration of hospitalization. The rate of in-hospital mortality was higher in group A (5%) vs. group B (0%), without statistical significance, *p* = 0.056. We observed significantly higher prevalence of 30-day cumulative MACCE among group A (26.6%) compared to group B (7%, *p* = 0.011). There were no significant differences between the two groups in ACS recurrence, hospitalization for heart failure, cardiovascular death or stroke at 30-day follow-up. We could not find any statistically significant associations between various predictors such as smoking, diabetes mellitus, culprit lesion, type of ACS or macrocytosis and 30-day MACCE (Figure 1). 

The present case-control study included the entire group of analyzed patients. All subjects were admitted with the diagnosis of acute coronary syndrome, with different types of presentation. One hundred patients from the study population were divided into two groups: group A—50 patients with HIV infection—and group B—50 patients without documented HIV infection (control group, matched by age and sex).

Data from studies reports that traditional cardiovascular risk factors are present with high prevalence in people living with HIV. In PLWH who have already developed manifest coronary artery disease, the rate of cardiovascular risk factors does not seem to differ significantly compared to controls. In our study, we found some differences compared to data from other studies in the literature. For instance, in our analysis, the prevalence of smoking in PLWH presenting with ACS was lower in comparison to the results from another similar study (63.3% vs. 79%) [27]. Systemic hypertension had significantly higher prevalence in our analysis compared to other studies (61.2% vs. 18–50%) [17,27,29]. Furthermore, dyslipidemia was present in 75% of our patients compared to 36.4–62.5% in other studies [17,27]. Although we found no differences in the overall prevalence of dyslipidemia between the two groups, people living with HIV had significantly lower values of total serum cholesterol compared to the control group (173.21 ± 50.79 mg/dL vs. 207.50 ± 62.16, *p* = 0.004). In our opinion, a plausible explanation is that lipid profile analysis in people living with HIV has become more frequent and systematic in the last few years, and statins are routinely recommended. Diabetes, an important risk factor for cardiovascular complications (regardless of the type or treatment) was more prevalent in the control group, but the difference had no statistical significance. In comparison to other studies, diabetes had higher prevalence in our patients (18.4% vs. 9–12.5%) [17,27]. Even though diabetes and obesity are commonly associated, in our study, obesity was found in a small number of PLWH with acute coronary syndrome compared to the control group (8.2% vs. 34%, *p* = 0.003), likely explained by the presence of HIV-infection-associated lipodystrophy and by the use of antiviral drugs such as stavudine, which are now avoided and have been replaced by other drugs from the nucleoside reverse transcriptase inhibitors (NRTIs) class [30,31]. Our finding is concordant with the results of another study that reported lower body mass indeces in PLWH with acute coronary syndrome compared to patients without HIV (22 ± 3.1 kg/m^2^ vs. 27 ± 4.7 kg/m^2^ respectively), suggesting that obesity is more frequently present in patients not infected with HIV.

Comparing the data found in the present study to the published data, people living with HIV who have associated acute coronary syndromes have a lower prevalence of traditional cardiovascular risk factors (dyslipidemia, systemic hypertension, obesity) compared to patients with acute coronary syndromes and no documented HIV infection [25]. In contrast to the current data, we found a lower prevalence of smoking in PLWH with acute coronary syndromes compared to patients with acute coronary syndromes without documented HIV infection [20,21,22,27], possibly because smoking has higher prevalence in the general Romanian population than in other countries.

### 3.6. Clinical Characteristics

Atypical angina as a main symptome was 10 times more common in PLWH than in the control group. It is well known that thoracic pain in PLWH with acute coronary syndromes may not present as typical angina, just as in patients with diabetes or chronic kidney disease [8,17]. Although many similar cases of atypical angina at presentation are described in the literature [18,32], there is no clear data regarding the prevalence. Previous coronary artery disease diagnosis was another aspect analyzed in our study. Our results showed that history of coronary artery disease was comparable between the two groups, with no statistically significant differences (30% in HIV-positive vs. 24% in HIV-negative patients), which is also similar to the presently availabe data (27–34%) [17,22]. However, peripheral arterial determinations of atherosclerosis (lower extremity artery disease, carotid artery disease or ischemic stroke) were significantly different between the two groups, with higher prevalence in PLWH with acute coronary syndromes (27.7%) compared to the control group (8%), *p* = 0.011.

### 3.7. Laboratory Findings

Laboratory biomarkers were analyzed in the entire group. Among the biomarkers of anemia, mean corpuscular volume, serum creatinine, creatinine clearance, serum lipids and myocardial necrosis markers (creatinkinase and creatinkinase-MB), we found significant differences for the categorical and continuous variables of mean corpuscular volume and for the continuous variables of creatinine clearance, total serum cholesterol and maximum serum values of total creatinkinase and its fraction, creatinkinase-MB. Frequent macrocytosis is described in studies in PLWH, as a consequence of exposure to zidovudine or stavudine [33]. One study that included 130 HIV-negative patients with interventional revascularization found that macrocytosis was associated with a worse prognosis, even in the absence of anemia [34]. In our study, the values of mean corpuscular volume (MCV) were higher in the group of PLWH (mean value of 95.32 [DS 9.92] fL) compared to the control group (mean value of 89.19 [DS 9.33] fL), a result with powerful statistical significance, *p* = 0.002. It is important to note that we found no data from studies related to macrocytosis in PLWH with acute coronary syndromes, an association that should perhaps be studied separately. Creatinine clearance was significantly lower in PLWH with acute coronary syndromes in our study. Kidney disease is becoming more common among PLWH for several reasons, including the increase in life expectancy and the increased prevalence of risk factors such hypertension or diabetes. Certain antiretroviral drugs can lead to renal dysfunction. The group of PLWH with acute coronary syndrome had lower values compared to patients with similar profiles from other studies (mean value of 85.5 mL/min/m^2^ vs. 130 mL/min/m^2^) [17]. Although the prevalence of dyslipidemia in both study groups was similar, total serum cholesterol was significantly lower in the PLWH group compared to the control group (mean value of 173.2 [DS 50] mg/dL vs 207.5 [DS 62.1] mg/dL, *p* = 0.004). We opine that this finding may be explained by the more frequent monitorization of the lipid profile and more frequent recommendation of statins in these patients in the last few years. In both groups, the mean values of total serum cholesterol were quite similar to those cited in the literature (174 ± 44 mg/dL vs. 185 ± 45 mg/dL, *p* = 0.44 [27]; 197 ± 63 mg/dL vs. 205 ± 59 mg/dL, *p* = 0.51 [22]); 207 ± 57 mg/dL vs. 203 ± 49 mg/dL, *p* = 0.92 [20]). Even though it is recognized that the prevalence of dyslipidemia is higher in PLWH than in the general population, this finding is not necessarily relevant to patients that associated an acute coronary syndrome in their evolution. Furthermore, data from previous studies do not indicate statistically significant differences, whereas in the present study, we found significant differences between both groups (*p* = 0.004). The presence of myocardial necrosis was analyzed only from the values of total serum creatinkinase (CK) and its fraction, creatinkinase-MB (CK-MB), because data concerning the values of serum troponines were relatively scarce in the patients’ files. Generally, we found significantly lower values of serum CK and CK-MB in PLWH with acute coronary syndromes compared to the control group (mean value of CK 715.95 U/L vs. 1485 U/L, *p* = 0.008 and a mean value of CK-MB 103.72 U/L vs. 157.06 U/L, *p* = 0.039). One possible cause for these results is that unstable angina was dominant in the group of PLWH with acute coronary syndromes. Our findings are concordant with the results of another study published in the literature that included 96 patients: 48 with acute coronary syndrome and HIV and 48 controls. The authors also reported higher maximum values of serum CK-MB in controls, although without statistical significance (128 ± 189 ng/mL vs. 160 ± 181 ng/mL, *p* = 0.3) [27].

We found that people living with HIV were less likely to present with ST elevation myocardial infarction than controls (42% vs. 74%, *p* = 0.012). Regarding coronary acute syndrome type, the literature showed different data. ST elevation myocardial infarction was the most frequent in a cohort of 44 HIV-infected patients studied by Perello et al. (59% vs. 24%), but other studies reported lower prevalence of ST elevation myocardial infarction (49% vs. 56% [20] and 54% vs. 62.5% [27]). As far as the coronary angiography features are concerned, our study showed that the features of the two groups were not greatly different with respect to culprit lesions. In our study, the most important differences regarding the culprit lesion referred to left main involvment, which had a higher prevalence among PLWH compared to controls, through without reaching a statistically significant difference (7% vs. 2%). The prevalence of left main coronary artery lesions in PLWH is around 6%, which is equivalent to that of the general population [35,36].

Compared to the uninfected patients, we found that PLWH had significantly higher prevalence of high SYNTAX I scores (≥23 points) (24% vs. 4.4%, *p* = 0.019). By analyzing the continuous variable of SYNTAX I score, we found significant differences between the two groups (13.45 [DS 11.82] points vs. 7.86 [DS 9.78] points, *p* = 0.022). A recent study showed an increased prevalence of significant coronary lesions among HIV patients; however, they had a similar rate of multivessel coronary artery disease, no significant differences of baseline SYNTAX score, but significantly higher residual SYNTAX score in PLWH [23]. 

Higher hospitalization duration has been reported among people living with HIV, compared to HIV-negative patients (8.12 [DS 11.07] days vs. 6.38 [DS 2.24] days), though without reaching a statistically significant difference (*p* = 0.279). A recent study showed equal hospitalization duration between people living with HIV and controls (5.7 ± 3.3 days vs. 5.7 ± 1.6 days, *p* = 1), a shorter period compared to our results. During hospitalization for acute coronary syndrome, studies that included an HIV-negative control group did not find significant differences in cardiovascular mortality [17,20,37,38]. People living with HIV had a higher in-hospital mortality rate compared to controls (10% vs. 0%, *p* = 0.056). The higher in-hospital mortality in people living with HIV could be explained by their more frequent history of coronary artery disease, their higher Killip severity class at admission, their more complex coronary disease and their greater number of comorbidities. People living with HIV have increased recurrence of episodes of acute coronary syndrome (8.9% vs. 7%), HF requiring hospitalization (8.9% vs. 0%), cardiovascular death (6.7% vs. 0%) and stroke (2.2% vs. 0%), though without reaching a statistically significant difference. We reported more cumulative 30-day major adverse cardiovascular events (MACCE) in people living with HIV compared to controls (26.6% vs. 7%, *p* = 0.021). Recent data has shown that the short-term follow-up, including in-hospital mortality and major adverse cardiac and cerebrovascular events, in people living with HIV is similar to those who are HIV-negative, but the risk of recurrent ischemic events and the rate of HF requiring hospitalization after an episode of ACS is higher in HIV-positive patients [27]. Other studies have shown that there are no differences between people living with HIV presenting with acute coronary syndrome and HIV-negative patients with acute coroanry syndrome in terms of occurrence of postinfarction angina or 30-day mortality [17]. Older studies showed that the hospitalization rate for coronary artery disease was significantly higher in people living with HIV compared to controls [16,39]. We used logistic regression analysis to evaluate the relationship between various predictors and 30-day MACCE. We could not find any statistically significant associations between various predictors, such as smoking, diabetes mellitus, culprit lesion, type of ACS or macrocytosis, and 30-day MACCE (Figure 1). The long-term prognosis after an ACS in PLWH is unclear, but it is certain that they have a worse cardiovascular risk profile [27]. A large study of 41-month follow-ups after coronary artery bypass graft revascularization found that people living with HIV had higher rates of adverse events compared to controls, mostly because of the need to repeat revascularization after using percutaneous coronary intervention of the native coronary arteries [28].

This study had several limitations: firstly, its retrospective nature; and secondly, its relatively small number of patients. In addition, the use of specific cardiovascular medications was not controlled during the study, and we were not able to assess the role of specific ART regimens in patients’ outcomes.

Although our study was performed on a relatively small number of patients, this is the only study to date performed in Romania with multicenter enrolment and a HIV-uninfected group matched for age and sex.

## 4. Conclusions

People living with HIV presenting with acute coronary syndrome can have various characteristics. Traditional and specific cardiovascular risk factors can lead to early, diffuse and accelerated atherosclerosis. The symptomatology of acute coronary syndromes in people living with HIV can be atypical, potentially creating diagnostic issues. The degree of coronary involvement, as assessed by a SYNTAX I score, can be higher than in controls. At short-term follow-up, people living with HIV have more 30-day major adverse cardiac and cerebrovascular events than HIV-negative patients.

## Figures and Tables

**Figure 1 diagnostics-13-01526-f001:**
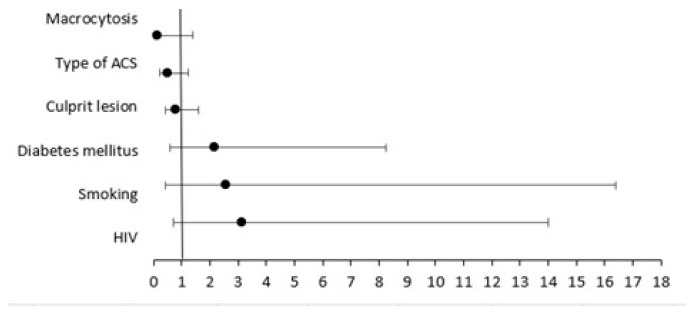
30-day major adverse cardiac and cerebrovascular events predictors.

**Table 1 diagnostics-13-01526-t001:** Epidemiological data, cardiovascular risk factors and comorbidities.

	Group A	Group B	*p*
**Epidemiological data** Male, *n* (%) Age, mean (SD)	47 (94%) 49 (9.03)	46 (92%) 49.62 (11.43)	
**Cardiovascular risk factors** Smoking, *n* (%) Hypertension, *n* (%) Dyslipidemia, *n* (%) Diabetes mellitus, *n* (%) Personal history of CAD, *n* (%) Obesity, *n* (%)	32 (64%) 30 (60%) 37 (74%) 9 (18%) 15 (30%) 4 (8%)	43 (86%) 32 (64%) 38 (76%) 12 (24%) 12 (24%) 17 (34%)	0.011 0.775 0.955 0.464 0.499 0.003
Atypical angina, *n* (%)	10 (20%)	1 (2%)	0.003
**Comorbidities** Anemia, *n* (%) Chronic kidney disease, *n* (%) Peripheral artery disease, *n* (%)	14 (28%) 9 (18%) 13 (26%)	7 (14%) 5 (10%) 4 (8%)	0.078 0.183 0.011

**Table 2 diagnostics-13-01526-t002:** Type of acute coronary syndrome, severity of acute myocardial infarction, angiographical features and SYNTAX I score.

	Group A	Group B	*p*
**Type of ACS (global)** STEMI, *n* (%) NSTEMI, *n* (%) UA, *n* (%)	21 (42%) 12 (24%) 17 (34%)	37 (74%) 5 (10%) 8 (16%)	0.012 0.001 0.006 0.037
**Culprit lesion (global)** LAD, *n* (%) LCX, *n* (%) RCA, *n* (%) LM, *n* (%) Other, *n* (%)	20 (46.5%) 7 (16.3%) 7 (16.3%) 3 (7%) 6 (14%)	20 (40%) 12 (24%) 16 (32%) 1 (2%) 2 (4%)	0.063 0.527 0.357 0.079 0.238 0.087
**Number of diseased vessels (global)** One-vessel, *n* (%) Two-vessel, *n* (%) Three-vessel, *n* (%)	20 (46.5%) 7 (16.3%) 7 (16.3%)	20 (40%) 12 (24%) 16 (32%)	0.025 0.527 0.357 0.079
**SYNTAX SCORE** SYNTAX I ≥23 points, *n* (%) SYNTAX I points, mean (SD)	9 (24.3%) 13.45 (11.82)	2 (4.4%) 7.86 (9.78)	0.019 0.022

**Table 3 diagnostics-13-01526-t003:** Biological Tests.

	Group A	Group B	*p*
Hemoglobin g/dL, mean (SD) MCV fL, mean (SD) Creatinine mg/dL, mean (SD) GFR (CKD-EPI) mL/min/m^2^, mean (SD) Total cholesterol mg/dL, mean (SD) Triglycerides mg/dL, mean (SD) CK peak u/L, mean (SD) CK-MB peak u/L, mean (SD)	14.09 (1.92) 95.32 (9.92) 1.23 (0.99) 85.51 (33.51) 173.21 (50.79) 189.02 (126.81) 715.95 (985.80) 103.72 (106.40)	14.59 (1.41) 89.19 (9.33) 1.04 (1.54) 101.79 (31.30) 207.50 (62.16) 172.46 (95.51) 1485.44 (1514.32) 157.06 (139.77)	0.141 0.002 0.457 0.014 0.004 0.473 0.008 0.039

**Table 4 diagnostics-13-01526-t004:** HIV infection classification of patients with ACS.

HIV Stage	*n* (%)
A1 A2 A3 B1 B2 B3 C1 C2 C3 no data	7 (14%) 2 (4%) - 12 (24%) 5 (10%) - 8 (16%) 10 (20%) 3 (6%) 3 (6%)

**Table 5 diagnostics-13-01526-t005:** Immune status assessment at presentation with ACS.

CD4+ Lymphocytes	*n* (%)
<200 cells/µL 200–500 cells/µL ≥500 cells/µL no data	3 (6%) 18 (36%) 27 (54%) 3 (6%)

**Table 6 diagnostics-13-01526-t006:** Antiretroviral exposure of patients before and at the moment of ACS.

Antiretroviral Therapy (ART) Exposure	At ACS Presentation	Previous
**NRTIs** Abacavir, *n* (%) Lamivudine, *n* (%) Zidovudine, *n* (%) Stavudine, *n* (%) Emtricitabine, *n* (%) Tenofovir alafenamide, *n* (%) Tenofovir disoproxil fumarate, *n* (%)	16 (32%) 23 (46%) 6 (12%) 0 (0%) 14 (28%) 6 (12%) 13 (26%)	13 (26%) 18 (36%) 13 (26%) 5 (10%) 0 (0%) 0 (0%) 1 (2%)
**NNRTs** Etravirine, *n* (%) Doravirine, *n* (%) Efavirenz, *n* (%)	3 (6%) 1 (2%) 4 (8%)	4 (8%) 0 (0%) 4 (8%)
**PIs** Darunavir, *n* (%) Lopinavir, *n* (%) Atazanavir, *n* (%) Ritonavir, *n* (%) Amprenavir, *n* (%) Saquinavir, *n* (%)	8 (16%) 5 (10%) 3 (6%) 17 (34%) 0 (0%) 0 (0%)	3 (6%) 10 (20%) 6 (12%) 14 (28%) 2 (4%) 8 (16%)
**IIs** Dolutegravir, *n* (%) Bictegravir, *n* (%) Raltegravir, *n* (%) Elvitegravir, *n* (%)	7 (14%) 3 (6%) 8 (16%) 2 (4%)	1 (2%) 0 (0%) 2 (4%) 1 (2%)
**Without ART**, *n* (%)	5 (10%)	-

**Table 7 diagnostics-13-01526-t007:** Short-term follow-up.

	Group A	Group B	*p*
**Short-term follow-up** Duration of hospitalization, mean (SD) In-hospital mortality, *n* (%) **30-day MACCEs** -Recurrent ACS, *n* (%) -HF requiring hospitalization, *n* (%) -Cardiovascular death, *n* (%) -Stroke, *n* (%)	8.12 (11.07) 5 (10%) 12 (26.6%) 4 (8.9%) 4 (8.9%) 3 (6.7%) 1 (2.2%)	6.38 (2.24) 0 (0%) 3 (7%) 3 (7%) 0 (0%) 0 (0%) 0 (0%)	0.279 0.056 0.021 1 0.117 0.242 1

## Data Availability

Data is unavailable.

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
