# Peer review of "Cardiovascular Risk Factors, Angiographical Features and Short-Term Prognosis of Acute Coronary Syndrome in People Living with Human Immunodeficiency Virus: Results of a Retrospective Observational Multicentric Romanian Study"

_diagnostics, 2023, doi:10.3390/diagnostics13091526_

Round 1

Reviewer 1 Report

Major comments.

  1. A statistical summary should be given to show that the number of patients in the groups is sufficient to draw conclusions about the frequency of different types of acute coronary syndrome (ACS) and the severity and localization of atherosclerotic lesions. 
  2. Given the significant prevalence of the disease, the control group should be formed not only by gender and age but also by the type of ACS and the localization of the atherosclerotic lesions.
  3. The laboratory data (CK-MB, creatinine) may be directly related to the incidence of ACS with ST-segment elevation. The authors do not discuss this issue.
  4. A major shortcoming is the lack of data on the status of patients with HIV at the time of onset of ACS. Were they receiving antiretroviral therapy? What kind of therapy? One could at least give the CD4+ lymphocyte count.
  5. Everything about prognosis is undoubtedly a highlight of the article. But this should be discussed in the context of the type of ACS, availability and type of specific therapy. The data should be compared with global national statistics on the prognosis of HIV-negative patients with any type of ACS.
  6.  

Minor comments.

  1. Lines 53-57 in the Introduction: "There are three key sequential biological processes that accelerate progression of atherosclerosis in PLWH: inflammation; 2- transformation of monocytes to macrophages and foam cell; 3-apotosis of foam cells leading to plaque development through Ca2+-dependent endoplasmic reticulum stress”. These pathogenetic mechanisms are not specific to HIV-associated atherosclerosis. 
  2. What are the implications of "geography" for the course of ACS in HIV-infected individuals?   
  3. Methods. Ethical committee approval is not mentioned.
  4.  How was the association between MACCE and risk factors (Figure.1) assessed?
  5.  The authors write: "Patients were analyzed for cardiovascular risk factors, medical history and therapy", but there is no data on therapy in the results.
  6. In the text, before the tables, it should be stated that group A is patients with HIV and group B is a control group.
  7. The authors write that patients with HIV often have symptoms of “atypical angina”. These symptoms should be described. 
  8. What does the p in table 2 refer to?
  9. Line 191. The error in the value of «total СК in PLWH 145.85±1514.32”. 
  10. The authors write: "creatinine clearance was significantly lower in PLHIV and in acute coronary syndrome". The authors should discuss what this might be due to. 
  11. Incorrect format of references (no volume and page numbers).

Author Response

Dear reviewer,

First of all, we want to thank you for your corrections and suggestions.

We will answer to your every single request.

Major comments.

  1. A statistical summary should be given to show that the number of patients in the groups is sufficient to draw conclusions about the frequency of different types of acute coronary syndrome (ACS) and the severity and localization of atherosclerotic lesions. 

Answer:  If you refer to the necessity to calculate a sample size estimation for the study (e.g. 80% power and alpha error 0.05), it is not applicable. This normally applies to prospective randomized interventional studies with large pool of possible applicants. In 2022, in Romania, there were 17700 patients with HIV. Having in mind that the average ACS incidence is around 180 cases per 100000 pts per year, (1-2) the combined incidence of ACS in HIV in 13 years should be around 400 pts. Also, the area covered by Bucharest and southern Romania consists of about 40% of population, so around 200-250 cases of ACS in 13 years. By all measures, the current study consists of 20-25% of the entire ACS and HIV population in the last 13 years, including those that did not report.

Just for an exercise, to make a study with 80% power, 0.05 alpha and an expected difference in MACCE incidence of 2%, we would have need a population of 13000. To conclude, there are very few patients with HIV and ACS, also many of the similar studies have similar populations.

If you refer to small/scarce numbers statistics, as I have reported, we used Fischer exact test where possible

  1. Given the significant prevalence of the disease, the control group should be formed not only by gender and age but also by the type of ACS and the localization of the atherosclerotic lesions.

Answer:  Unfortunately, when the statistics were made, it was decided that the control group would only be matched for age and gender.

  1. The laboratory data (CK-MB, creatinine) may be directly related to the incidence of ACS with ST-segment elevation. The authors do not discuss this issue.

Answer: We have written in the main text that one possible cause for lower CK-MB and creatine kinase values in group A can be the absence of myocardial necrosis in unstable angina.

  1. A major shortcoming is the lack of data on the status of patients with HIV at the time of onset of ACS. Were they receiving antiretroviral therapy? What kind of therapy? One could at least give the CD4+ lymphocyte count.

Answer: We have data related to the stage of HIV infection, antiretroviral therapy at the time of the coronary syndrome and the number of CD4+ lymphocyte dosed very close to the moment of ACS. We made a table with all this data and we will attach it on the manuscript.

  1. Everything about prognosis is undoubtedly a highlight of the article. But this should be discussed in the context of the type of ACS, availability and type of specific therapy. The data should be compared with global national statistics on the prognosis of HIV-negative patients with any type of ACS.

Answer: In Romania, there is no global nation statistics for ACS, for this reason we decided to make a case-control group.

Minor comments.

  1. Lines 53-57 in the Introduction: "There are three key sequential biological processes that accelerate progression of atherosclerosis in PLWH: inflammation; 2- transformation of monocytes to macrophages and foam cell; 3-apotosis of foam cells leading to plaque development through Ca2+-dependent endoplasmic reticulum stress”. These pathogenetic mechanisms are not specific to HIV-associated atherosclerosis. 

Answer: We took this information from an article published in an AHA journal series, Aterioscler Thromb Vasc Biol (link: https://www.ahajournals.org/doi/full/10.1161/ATVBAHA.113.302191 ). The authors probably meant that these three mechanisms are accelerated in HIV patients. If it doesn’t seem right to you, we can eliminate this paragraph.

  1. What are the implications of "geography" for the course of ACS in HIV-infected individuals?

Answer: By geographical differences, we wanted to emphasize the fact that the cause of death in PLWH differs according to the region where they are from(in developed countries, coronary artery disease had become one of the most important causes of death while in Sub-Saharan countries the mortality is still linked to infections). We removed this paragraph so that we would not confuse the readers.

  1. Methods. Ethical committee approval is not mentioned.

Answer: We mentioned the ethical committee approval in the  “Institutional Review Board Statement” section. We will also mention it in the methods.

  1. How was the association between MACCE and risk factors (Figure.1) assessed?

Answer: We used logistic regression analysis to evaluate the relationship between various predictors of 30-day MACCE and we could not find any statistically significant associations between various predictors and MACCE.

  1.  The authors write: "Patients were analyzed for cardiovascular risk factors, medical history and therapy", but there is no data on therapy in the results.

Answer: We will also provide details about antiretroviral therapy in a table.

  1. In the text, before the tables, it should be stated that group A is patients with HIV and group B is a control group.

Answer: We will mention in the “Materials and Methods” section.

  1. The authors write that patients with HIV often have symptoms of “atypical angina”. These symptoms should be described. 

Answer: We will mention in the text the fact that we consider atypical angina pain that does not have the characteristics of angina or dyspnea as ACS presentation.

  1. What does the p in table 2 refer to?

Answer: p in table 2 is “global p” and refers to the differences between the two groups calculated by chi-square test calculator with contingency table 3x2, 5x2 and 3x2. We will also provide p value calculated for each item, with contingency table 2x2.

  1. Line 191. The error in the value of «total СК in PLWH 145.85±1514.32”. 

Answer: There was a mistake when writing in main text. The normal value was 1485.44±1514.32.

  1. The authors write: "creatinine clearance was significantly lower in PLHIV and in acute coronary syndrome". The authors should discuss what this might be due to. 

Answer: Renal dysfunction in PLWH is a very complex subject. We will write some possible mechanisms of renal dysfunction in these patients.

  1. Incorrect format of references (no volume and page numbers).

Answer: We will complete the references with the volume, page numbers and also with DOI.

We hope that the answers are adequate and the changes make the manuscript better. Thank you for your review.

Reviewer 2 Report

It is necessary to describe in more detail the inclusion and exclusion criteria for the selection of the patients in the methodology section. It is redundant to repeat the results through tables and text. There are discrepancies in the percentage values ​​for the listed comorbidity anemia in the text (line 157) (10% vs. 2%, p=0.078) and Table 1 (28% vs. 14%, p=0.078).

Author Response

Dear reviewer,

First of all, we want to thank you for your corrections and suggestions.

We will answer to your every single request.

It is necessary to describe in more detail the inclusion and exclusion criteria for the selection of the patients in the methodology section. It is redundant to repeat the results through tables and text. There are discrepancies in the percentage values ​​for the listed comorbidity anemia in the text (line 157) (10% vs. 2%, p=0.078) and Table 1 (28% vs. 14%, p=0.078)

Answer:

  • We will write the inclusion and exclusion criteria.
  • We wrote the results from the table in the main text to make it easier for the reader to follow. We will write only those findings with statistical significance in the text.
  • We will correct the discrepancies in the percentage values from the text and from the table. We also observed, that not only the values of anemia are different, but also the percentage values for the prevalence of vessel disease.

We hope that the answers are adequate and the changes make the manuscript better. Thank you for your review.

Round 2

Reviewer 1 Report

In my opinion, the authors have substantially improved the article, and all aspects that could be corrected have been corrected.

 Minor revision is still needed.

The section on statistics (methods) should be expanded to mention the use of logistic regression analysis in assessing the relationship between various predictors and MACCE.

 The authors write in the discussion that statins are prescribed to HIV+ patients to prevent the development of hypercholesterolemia. The question arises whether the group A and B patients were taking statins. If this information is available, it should be added to the Results section.

I would like to clarify a controversial fragment in the introduction ("There are three key sequential biological processes that accelerate the progression of atherosclerosis in PLWH: 1 - inflammation; 2- transformation of monocytes to macrophages and foam cell; 3-apoptosis of foam cells leading to plaque development through Ca2+-dependent endoplasmic reticulum stress").  Indeed, it should be stressed here that these common pathogenetic mechanisms are accelerated in HIV patients. In paragraph number 3, I think the logic of events is broken, I would have written "Ca2+-dependent endoplasmic reticulum stress and apoptosis of foam cells leading to plaque development".  But this is not a point of principle.

Author Response

Dear reviewer,

Thank you for your suggestions.

We will answer to you every single request.

The section on statistics (methods) should be expanded to mention the use of logistic regression analysis in assessing the relationship between various predictors and MACCE.

Answer: We will mention this thing.

The authors write in the discussion that statins are prescribed to HIV+ patients to prevent the development of hypercholesterolemia. The question arises whether the group A and B patients were taking statins. If this information is available, it should be added to the Results section.

Answer: Unfortunately, we don’t have enough data on statins use before ACS.

I would like to clarify a controversial fragment in the introduction ("There are three key sequential biological processes that accelerate the progression of atherosclerosis in PLWH: 1 - inflammation; 2- transformation of monocytes to macrophages and foam cell; 3-apoptosis of foam cells leading to plaque development through Ca2+-dependent endoplasmic reticulum stress").  Indeed, it should be stressed here that these common pathogenetic mechanisms are accelerated in HIV patients. In paragraph number 3, I think the logic of events is broken, I would have written "Ca2+-dependent endoplasmic reticulum stress and apoptosis of foam cells leading to plaque development".  But this is not a point of principle.

Answer: We modified the paragraph as you suggested. If it is still confusing for the future readers, we can remove it.